# Spider Silk-Improved Quartz-Enhanced Conductance Spectroscopy for Medical Mask Humidity Sensing

**DOI:** 10.3390/molecules27134320

**Published:** 2022-07-05

**Authors:** Leqing Lin, Yu Zhong, Haoyang Lin, Chenglong Wang, Zhifei Yang, Qian Wu, Di Zhang, Wenguo Zhu, Yongchun Zhong, Yuwei Pan, Jianhui Yu, Huadan Zheng

**Affiliations:** 1Key Laboratory of Optoelectronic Information and Sensing Technologies of Guangdong Higher Education Institutes, Department of Optoelectronic Engineering, Jinan University, Guangzhou 510632, China; llq119@stu2019.jnu.edu.cn (L.L.); z2769797220y@hotmail.com (Y.Z.); linhaoyang12@gmail.com (H.L.); wonderwcl@163.com (C.W.); braggyang0101@163.com (Z.Y.); wqianjnu@163.com (Q.W.); zhuwg88@jnu.edu.cn (W.Z.); ychzhong@163.com (Y.Z.); 2College of Traditional Chinese Medicine, Jinan University, Guangzhou 510630, China; zhangdi0915@126.com; 3Department of Preventive Treatment of Disease, The Affiliated TCM Hospital of Guangzhou Medical University, Guangzhou 510405, China; pywyvette@foxmail.com

**Keywords:** quartz tuning fork, quartz-enhanced photoacoustic spectroscopy, humidity sensor

## Abstract

Spider silk is one of the hottest biomaterials researched currently, due to its excellent mechanical properties. This work reports a novel humidity sensing platform based on a spider silk-modified quartz tuning fork (SSM-QTF). Since spider silk is a kind of natural moisture-sensitive material, it does not demand additional sensitization. Quartz-enhanced conductance spectroscopy (QECS) was combined with the SSM-QTF to access humidity sensing sensitively. The results indicate that the resonance frequency of the SSM-QTF decreased monotonously with the ambient humidity. The detection sensitivity of the proposed SSM-QTF sensor was 12.7 ppm at 1 min. The SSM-QTF sensor showed good linearity of ~0.99. Using this sensor, we successfully measured the humidity of disposable medical masks for different periods of wearing time. The results showed that even a 20 min wearing time can lead to a >70% humidity in the mask enclosed space. It is suggested that a disposable medical mask should be changed <2 h.

## 1. Introduction

Spider silk is well known for its outstanding mechanical properties. One of them is that spider silk is particularly sensitive to water [1]. When wetted or saturated in a high relative humidity (RH) atmosphere, spider silk ‘supercontracts’—unrestrained silk will shrink as much as 50% in length. The restrained silk generates stresses in excess of 50 MPa, which was discovered by Bell et al. and Guinea et al. [2,3]. The powerful water-collecting ability of spider silk is attributed to a unique fibrous structure consisting of periodic spindles and joints. The joints are composed of randomly scrambled, but neatly arranged nanofibers [4,5]. When the spider silk transitions from dry to wet conditions, the structure of spider silk will change to a joint to store water [6]. Water infiltrates the silk, disrupting the hydrogen bonding within the amorphous region of the proteins, which increases the molecular mobility to rearrange configurations [7,8,9,10]. This rearrangement results in the phenomenon of supercontraction that will change the modulus of the spider silk [11,12]. The spider silks are widely used in optical systems [13], the synthesis of new materials [14,15], biomedical applications [16,17,18,19,20,21] and tensile mechanics [22,23,24,25,26]. Spider silk also exhibits powerful cyclic contractions, which can be reversible and have reproducible use [9,27].

Quartz tuning forks (QTFs) have been traditionally used as a timing reference in wristwatches. The QTF is a bimorph cantilever based on the piezoelectric properties of the quartz [28]. The sensor consists of two prongs attached to a base, which is normally clamped to a holder. In order to make the mechanical movement of the fork tip work, an electrical field needs to be applied to the tuning fork tips, which is achieved by electroplating electrodes on the surface of the quartz [29]. Common materials attached to the QTF to improve the sensitivity include polystyrene (PS) nanofilm, polymethylmethacrylate (PMMA) nanowire, electrospun nanoporous wire, and a hydrogel conical tip, etc. [30,31,32,33,34]. The QTF’s remarkable advantages attract the attention of researchers, including its stable oscillation frequency, high quality factor, low power consumption, concise structure, and resistance to surrounding electromagnetic interference [35,36]. QTFs can be used in a wide temperature range from −40 °C to 85 °C and in an ultrahigh humidity environment [37]. Until now, QTFs have been used for micro force sensing, electric field intensity distribution detecting, charge distribution measurement, trace gas detection, polymer mechanical analysis, chemical/physical analysis and biosensing applications etc. [38,39,40,41,42,43,44,45,46,47,48]. A QTF sensor can be used for quantitative measurement of biomolecular interactions [49] or be used as an immunosensor for phenylketonuria diagnosis [50].

In this work, a sensitive and cost-effective humidity sensor was developed by using quartz-enhanced conductance spectroscopy (QECS) [51]. Compared with the spider silks as optical fibers for sensing applications, the QECS does not need to use optical instruments such as an optical spectrum analyzer for detection [25,26,52,53,54,55]. There is no requirement on the optical transmission of spider silk. The QECS has a quite remarkable advantage in self-sensing, since the results can be directly read out by means of electrical conductance spectroscopy and processed by a computer [51]. Compared to cantilevers or string resonators, there are no optics required for the QECS. Therefore, it will not introduce the thermoelastic noise and additional noise caused by the instability of the laser [56,57,58,59]. Due to the supercontraction ability of spider silk, the QTF was modified by natural spider silks to improve its humidity transducing ability. The developed humidity sensor was evaluated in different humid conditions, achieving a sensitivity of 12.7 ppm at 1 min. As proof of concept, the SSM-QTF sensor was used to measure the air humidity inside a medical mask to provide suggestions for mask wearing tips.

## 2. Experimental Setup for the Humidity Sensing System

QTFs were purchased from the Shenzhen XinChuangYue Electronic components Factory, with a resonance frequency of ~32.7 kHz and a Q factor of >60,000 in a sealed vacuum metallic package and ~10,000 in the air after removal of the metallic housing. The geometrical parameters were measured by an optical stereomicroscope. The prong length, prong width, and prong spacing of a QTF were 3.8 mm, 0.6 mm, and 0.3 mm, respectively. The effective spring constant of the QTF is ~20 kPa [48]. The effective elastic modulus of wet spider silk is ~17 GPa [60]. The detailed analysis of the structure and chemistry of spider silk can be found in references [61,62]. To attach spider silk onto a QTF, the metallic housing of a 3 × 8 mm QTF crystal was removed with two parallel prongs exposed. We obtained the spider silk from a Pterinochilus Murinus spider, which was fed in the laboratory. The spider silk was selected from the ampullate silk glands of a Pterinochilus Murinus spider. The spider silks are composed of tubiliform fibers, which are more sensitive to ambient humidity [47]. Without any sample pretreatment, the spider silk was bridged across the prongs of a QTF, forming a spider silk-modified QTF (SSM-QTF), as shown in Figure 1. A small amount of epoxy resin glue was used to glue the spider silk across the QTF prongs. The whole process was monitored by means of an optical microscope. The diameter of the microwire across the QTF prongs in Figure 1 was measured to be ~6 μm. Photographs of an SSM-QTF were taken with a Zeiss optical microscope.

To evaluate the humidity sensing performance of the SSM-QTF, we employed the experimental setup shown in Figure 2. The gas flow rate in the system was set to 20 standard cubic centimeters per minute (SCCM) by the mass flow controller (Alicat, Tucson, AZ, USA) to avoid gas flow noise. A pressure controller, pump, and valves were used to keep the pressure constant. The humidity in the chamber was controlled by a humidifier (Perma Pure, Lakewood, NJ, USA). In the gas stream, a commercial humidity sensor (Benetech GM1363B, Shenzhen, China) was also included for reference. The function generator (Tektronix AFG3102, Beverton, Oregon, USA) was used to produce sine waves with frequencies ranging from 32,500 Hz to 32,560 Hz with a fixed peak-to-peak amplitude of 400 mV. The SSM-QTF output electrical signals passed to a custom-made transimpedance preamplifier with a feedback resistance of 10 MΩ and were finally demodulated by a lock-in amplifier (SR830, Sunnyvale, CA, USA). The filter slope and time constant of the lock-in amplifier were set to 12 dB/Oct and 1 s, respectively. The demodulated signal was used to analyze the resonance QTF frequency, which varies with humidity. All experiments in this work were carried out at room temperature and atmospheric pressure. A LabVIEW program (Community Edition, National Instrument, Austin, TX, USA) was used to control the system, and all the measurements were performed automatically.

The resonance frequency *f* of the QTF can be expressed as [44,51]:(1)$$f_{QTF}=\frac{1}{2\pi }\times \sqrt{\frac{k_{QTF}}{m_{QTF}}},\;k_{QTF}=E_{QTF}\frac{wt^{3}}{4l^{3}}$$
where the *k_QTF_*, *m_QTF_*, and *E_QTF_* are the spring constant, effective mass, and Young’s modulus, respectively. The *t*, *l*, and *w* are the QTF prong thickness, QTF prong length, and QTF prong width, respectively. We considered that the mass of the spider silk was negligible, i.e., *dm_QTF_*≪*m_QTF_*. The resonance frequency shift, after modification by the spider silk, can be expressed as:(2)$$df_{QTF}=\frac{f_{QTF}}{2}\frac{dk_{QTF}}{k_{QTF}}$$

The resonance frequency shift can be attributed to the *dk_QTF_* value, which results from the spider silk spring constant *k*_silk_. Due to the fact that the QTF and the spider silk were connected in parallel, the effective spring constant *k_eff_* of the SSM-QTF can be expressed as:(3)$$E_{eff}\propto k_{eff}=k_{QTF}+k_{silk}$$

The electrical parameters of the SSM-QTF were measured by the abovementioned system in Figure 2, which is the same system as described in our previous publication [63]. The output of the conductance spectra was 1 spectrum/min, which was limited by the frequency scanning rate. The resonance frequency and Q factor of the SSM-QTF were calculated from a Lorenz curve fitting to the square of the amplitude associated with the conductance spectra [64].

## 3. Characterization of the Spider Silk-Modified Quartz Tuning Fork

A bare QTF without modification was also evaluated, under the same experimental conditions. The frequencies of the sine signals were scanned from 32,500 Hz to 32,560 Hz and from 32,725 Hz to 32,805 Hz with a step of 0.3 Hz for SSM-QTF and standard QTF, respectively. The resonance curves of the bare QTF and SSM-QTF are shown in Figure 3. In this work, we normalized the amplitude and focused on the frequency shift in the experiment. In the normalization, the amplitude of the highest signal was set as 1. Compared with a bare QTF, the resonance frequency peak of the SSM-QTF decreased from 32,764.9 Hz to 32,527.6 Hz, yielding a frequency shift of Δ*f* = 241.3 Hz. This can be attributed to the mass increase caused by the spider silk and the glue.

## 4. Spider Silk-Based Tuning Fork for Humidity Sensing

Figure 4a shows the conductance spectra changed in the SSM-QTF with humidity. The Lorentz function was used to fit the resonance curve to obtain the QTF resonance frequency and Q factor. As shown in Figure 4b, the resonance frequency decreased monotonically from 32 541.80 Hz at ~25% RH to 32 514.85 Hz at ~80% RH, with a Δ*f* = 25.95 Hz. This can be attributed to the modulus of spider silk being inversely proportional to the humidity; the effective modulus of the SSM-QTF decreased with the increase in the humidity [6]. The curve of Figure 4b was fitted using linear functions. An R square value of 0.99 was obtained. We also measured the frequency response of a bare QTF without the spider silk modification. No significant change was observed. The results are shown in the inset of Figure 4b, indicating that the large frequency shift Δ*f* was caused by the spider silk, not the QTF itself.

Figure 4c shows the *Q* factors of the SSM-QTF with different humidities. As the humidity increased, the spider silk absorbs water molecules, increasing the additional mass attached to the QTF. In addition, as the humidity increased, the damping effect by the water absorption into spider silk increased; thus, the quality factor of the SSM-QTF decreased [65]. As a result, with the relative humidity increases from ~25% RH to ~80% RH, the *Q* factor decreased from 5253 to 2790, by >2400. For comparison, the Q factor change of a bare QTF without spider silk modification was <200, shown in the inset of Figure 4c. The quality factor of SSM-QTF decreased monotonically but nonlinearly. Therefore, in this work, we used the frequency as the function of humidity to develop the sensor. Since the resonance frequencies and quality factors were obtained after the QTF reaches a saturated steady state, no estimation was necessary for the saturation of the SSM-QTF surface. This implies that the effect of water adsorption in the modified spider silk area dominates that in the quartz surface.

The relative humidity (RH) was converted into absolute humidity by the Vaisala humidity calculator. With an absolute humidity change from 8042 ppm to 24,056 ppm, the detection sensitivity of the SSM-QTF was calculated to be ~12.7 ppm. This humidity dynamic range was limited by our gas stream system, which can be improved by a better humidifier. The comparison of the SSM-QTF with current technology such as optical fibers, cantilever, and string resonators are summarized in Table 1. Although the sensitivity of the SSM-QTF sensor was not the highest, there are no optics required in the developed sensor. Compared to current technologies, this work is cost-effective with a high performance.

## 5. Stability of Spider Silk-Modified QTF Sensor

The resonance frequency of the SSM-QTF was measured with a fixed relative humidity. The humidity in the gas chamber was well controlled by using a mass flow controller (Alicat Tucson, AZ, USA) with a relatively low flow rate but higher resolution. With a relative humidity of 20 ± 0.1% RH, a resonance frequency of 32,542.88 Hz was obtained. Since the relative humidity is sensitive to temperature, the ambient temperature condition has to be properly maintained throughout the experiment. For the tested humidity range, the temperature in the humidity chamber has been monitored to be ~24 °C. It can be seen from Figure 5 that the SSM-QTF sensor has good stability over long-term running. Allan deviation was performed to evaluate the long-term stability of the developed sensor. The minimum deviation of the SSM-QTF resonance frequency was found to be 1.316 × 10^−3^ Hz at the optimum time of ∼30 min. With the optimal integration time, the detection sensitivity can be improved to ~0.8 ppm.

## 6. Humidity Evaluation of Medical Masks

The COVID-19 pandemic has forced the global population to adopt new ways of living, including the possible use of masks. Research has been conducted to evaluate the potential consequences of wearing masks for long periods [66,67]. Among various research, the high humidity in the mask for a long wearing time can lead to excessive moisture to the skin, disrupting the skin barrier and causing sensitive skin [68,69]. Part of the moisture is the condensation of the breathed air. The other part is the sweat that does not evaporate as quickly. If the skin is exposed to moisture for a long time, the barrier of our keratin will be undermined [70]. When the skin barrier is broken, the ability to prevent the loss of water in the skin and the ability to block external stimuli are reduced. [71]. Furthermore, wearing a mask for a long time will cause high humidity in the air inside the mask, which will easily cause the breeding of various pathogenic bacteria and increase the risk of skin diseases [72,73]. For some patients with seborrheic dermatitis, Malassezia can multiply on the skin surface when wearing a medical mask for a long time [74].

In the measurement, the humidity inside a disposable medical mask made of polypropylene was evaluated. The disposable medical masks were worn by the author Leqing Lin for different periods; then, they were taken off quickly to cover the SSM-QTF sensor in a sealed space. The humidity of the covered air inside the medical mask was then measured. For each measurement, the medical mask was replaced by a new one. The time clock was restarted. Except for the wearing time, all the other conditions were the same in the repeated experiments. The enclosed spaces were set up to prevent humidity interference from the outside atmosphere. Resonance frequencies of the different wearing time of the masks were recorded by a LabVIEW program. The obtained results were shown in Figure 6. Even a 20 min wearing time resulted in an air humidity >70% RH in the mask covered space. The longer the mask was worn, the greater the humidity in the mask. With a 150 min wearing time, the estimated humidity reached >78%. In fact, the humidity measured in the mask covered space was lower than the actual humidity surrounding the human mouth. In consequence, disposable medical masks are not suitable for prolonged wearing. It is suggested to change the mask within 2 h, when the relative humidity on the surface of the mask is more than 75%. For professional masks, such as the KN95 medical masks, the case was estimated to be worse due to their better “protection”.

## 7. Conclusions

This work featured a novel humidity sensing platform based on a quartz tuning fork (QTF) with a section of natural spider silk. The experimental results showed that the higher the environmental humidity, the lower the resonance frequency of the spider silk-modified QTF. The variation of the resonance frequency of the QTF modified by spider silk was closely correlated with the change in the environmental humidity. The evaluation was conducted in relative humidity from ~25% RH to ~80% RH. The resonance curves were recorded to obtain the SSM-QTF frequency variations by Lorentz fitting. With a 1 min working time, the detection sensitivity of ~12.7 ppm was obtained.

As proof of concept, the humidities of disposable medical masks, worn for different periods, were evaluated. The results suggested that the masks should be changed every 2 h due to excessive humidity, which may cause potential facial skin discomfort. Currently, the acquisition time for the resonance frequency and Q factor requires ~1 min. This time can be shortened by narrowing the frequency scanning time and range. Allan deviation was measured over a long time with a fixed humidity. With 30 points of integration, the detection sensitivity can be improved to ~0.8 ppm. Future work will include the development of a wearable respiration sensor to measure the humidity inside a medical mask. Compared to the current electrochemistry sensor [75,76], the SSM-QTF sensor has the advantage of being cost-effective and environmentally friendly. In addition, temperature effects will be researched. The temperature and humidity may be monitored simultaneously with the help of a machine learning method.

## Figures and Tables

**Figure 1 molecules-27-04320-f001:**
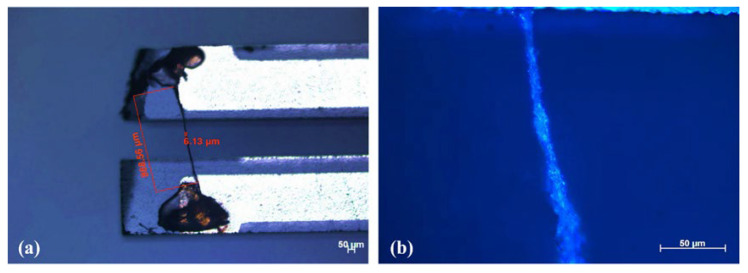
Photographs of an SSM-QTF taken with a Zeiss optical microscope. (**b**) is the enlarged view of (**a**).

**Figure 2 molecules-27-04320-f002:**
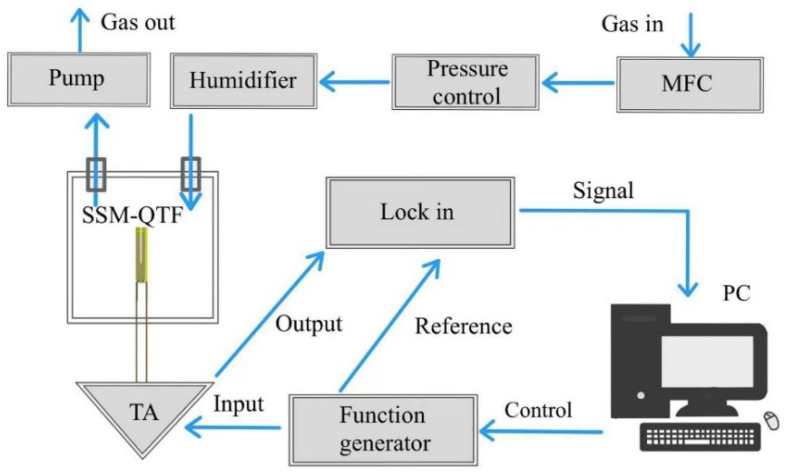
Experimental setup for humidity sensing. SSM-QTF: spider silk-modified QTF; TA: transimpedance amplifier; PC: personal computer; Lock-in: lock-in amplifier, MFC: mass flow controller.

**Figure 3 molecules-27-04320-f003:**
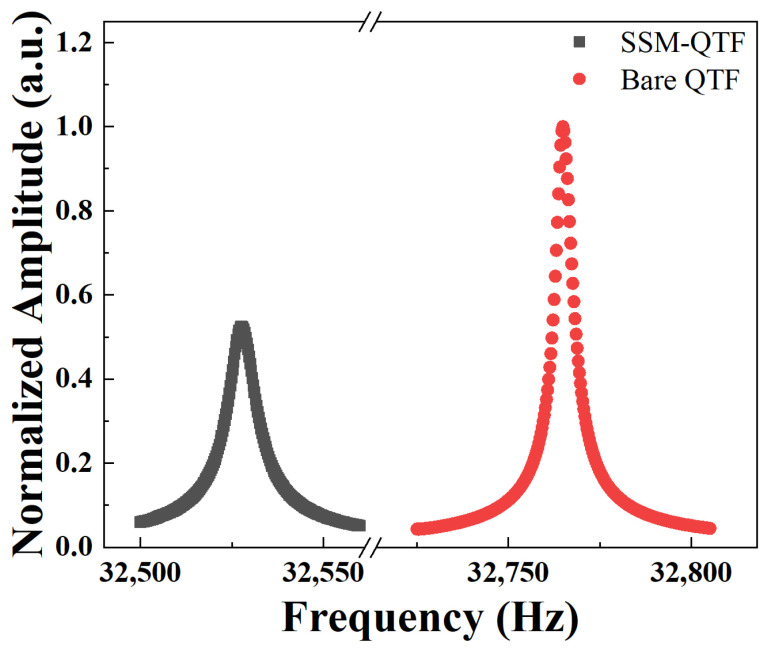
Conductance spectrum of a bare QTF without modification (red rounds) and after (black squares) modification by the spider silk.

**Figure 4 molecules-27-04320-f004:**
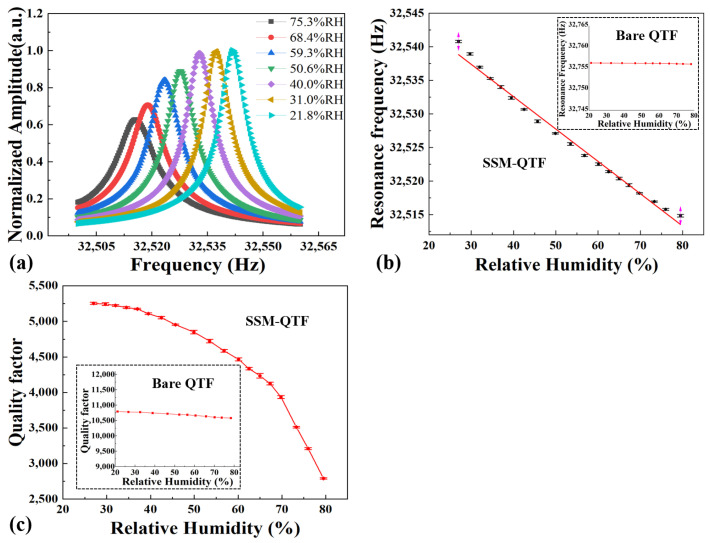
(**a**) Resonance curves of the SSM-QTF at different humidities. (**b**) Frequency variations of an SSM-QTF as a function of humidity. The figure inset shows the resonance frequency variations of a bare QTF. (**c**) Q factor variations of an SSM-QTF as a function of humidity. The figure inset shows the resonance frequency variations of a bare QTF.

**Figure 5 molecules-27-04320-f005:**
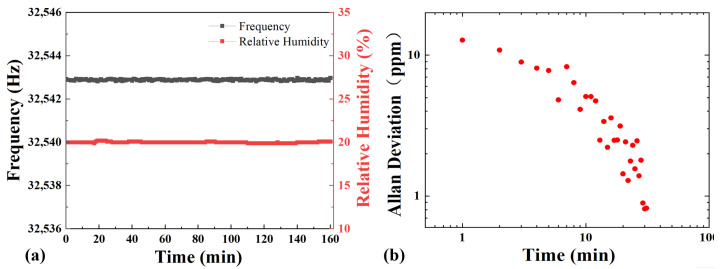
(**a**) Resonance frequency of the SSM-QTF and relative humidity measured for 160 min with the relative humidity controlled to be ~20% RH. (**b**) Allan deviations obtained by the SSM-QTF-based QECS systems.

**Figure 6 molecules-27-04320-f006:**
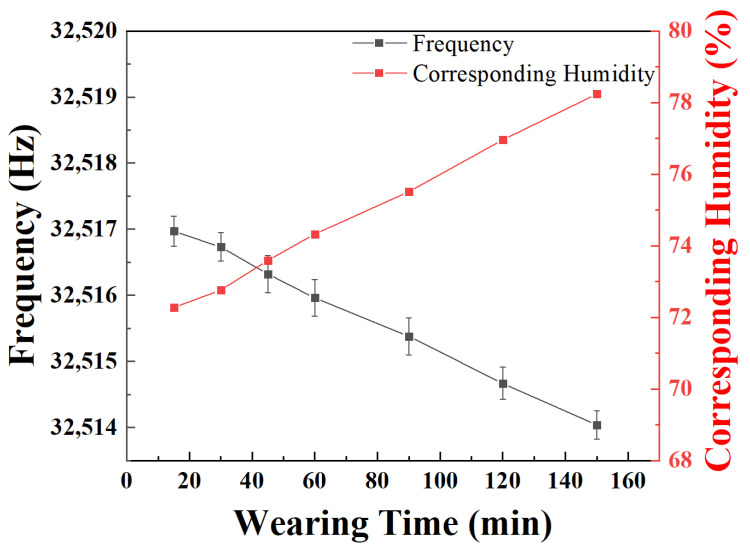
Resonance frequency variations of the SSM-QTF of the humidity inside the mask for different wearing times.

**Table 1 molecules-27-04320-t001:** Comparison of the SSM-QTF with optical fibers, cantilever, and string resonators.

Configuration	Sensitivity	Optics
optical fiber [12]	532 ppm/%RH	super-continuum laser
cantilever [26]	924 ppm/%RH	laser doppler vibrometer
string resonator [56]	2950 ppm/%RH	laser doppler vibrometer
SSM-QTF sensor	617 ppm/%RH	none

## Data Availability

Not applicable.

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
