# Peer review of "Spider Silk-Improved Quartz-Enhanced Conductance Spectroscopy for Medical Mask Humidity Sensing"

_molecules, 2022, doi:10.3390/molecules27134320_

Round 1

Reviewer 1 Report

The author describes an experimental system for developing a humidity sensing platform based on a spider silk modified quartz tuning fork. The experimental system seems promising but the author should address the following concerns:

1-      The title is confusing to me. What is the improved here? Why the author called it enhanced spectroscopy? Is this paper only reporting measuring humidity of medical masks?

2-       In the introduction part, the authors should discuss other applications of QTF (i.e. chemical and biological applications) which have been increasingly reported in the literature. Such discussion is very helpful to show the great potential of QTFs as sensitive sensing platforms.

3-      The following statement is not correct as there are other applications of QTFs. Please revise it.

Up to now, QTFs have been used for micro force sensing, electric field 46

intensity distribution detecting, charge distribution measurement of a single atom, trace 47gas detection, and polymer mechanical analysis [18-24].

4-  The introduction part should compare the current technology (QTF) to other existing technologies in which Spider silks were used (Ex:  spider silks were reported as optical fibers for sensing applications).

5-      The statement (from line 50 to 53) is almost identical to the statement published in a previous work (Appl. Phys. Lett. 107, 221903 (2015); https://doi.org/10.1063/1.4936648)

6-      The following statement is not clear and need to be –rewritten.

For application, the air humidity of 57surrounding a medical mask was measured, which gives a suggestion of mask wearing to 58 people.

7-      Sentences in lines 104,105 and 106 are structurally incorrect and need to be revised.

8-      A graph showing the change of quality factor as a function of humidity should be added

9-      The authors should explain how was the output signal is plotted as a normalized amplitude.

10-   The entire manuscript should be checked again for grammatical mistakes and sentence structures (also spacing between words)

Author Response

Many thanks for the reviewer's great improvement on the manuscript. Please find in the attachment the response letter.

Reviewer 2 Report

In the manuscript entitled “Spider silk improved quartz-enhanced conductance spectroscopy for medical mask humidity sensing, L. Lin et al. have reported a humidity sensing platform based on spider silk modified quartz tuning fork.

In the introduction, the authors should indicate the advantages and disadvantage of using spider silk, specifying the application technology, and reporting references. Moreover, the authors should also indicate the common technologies used for medical mask humidity sensing, reporting their advantages and disadvantages.

In the introduction, the authors should briefly describe QTFs technology and the common materials used to make it.

In the introduction, the authors should describe the QECS, specifically for humidity sensing, adding a table in which compare the advantage and the disadvantages of cantilevers and string resonators, reported in literature.

In the experimental section, the authors should add more details on the used spider silk, specifically how it was produced, selected and fixed. A chemical, structural and morphological characterization of the spider silk is necessary. To prepare the SSM-QTF a single filament of silk has been used, a SEM image or a better optical microscope image a necessary to evidence not only the length but also the diameter dimension.

All definitions and equations should be added in the experimental section.

Please, compare the reported results with those reported in literature with other materials or technology.

The authors should evaluate the temperature effect on the sensor response; moreover, which are the experimental conditions used to evaluate the sensor response? The authors should also consider the presence of not only air but also carbon dioxide during the measurement, simulating the real conditions. Please, discuss.

Please, describe how the measurement of the humidity inside the mask at different wearing time were conducted.

Please, try to explain, from a physical and chemical point of view, the improvement of the conductance spectrum using spider silk, adding references.

Please, add the error bar of values reported in all plots.

The English style is quite acceptable.

I can accept with major revisions.

Author Response

Many thanks for the reviewer's great improvement on this manuscript. Please find the response letter and revised manufacture in the attachment.

Round 2

Reviewer 1 Report

I thank the authors for improving the manuscript. I accept the manuscript after making the following minor corrections:

11)   Line (92): Photograph of an SSM-QTF 92 taken with a Zeiss optical microscope.

The sentence is incomplete. Needs to be completed

22)   The caption of Figure 1 should include Fig 1 (a) and Fig 1( b) NOT Fig 1 and Fig b

33)   Figures are sometimes written within the text as Figures and sometimes as Fig. Please be consistent

44)   The comparison table should not be added in the conclusion. It should be presented in the discussion section 

Author Response

Many thanks for the great improvement on the manuscript. Please see the revisions in the attachment.

Reviewer 2 Report

The revised version of the manuscript is now acceptable for publication.

Author Response

Thanks again for the great improvement on the manuscript.